# Factored Bandits

**Julian Zimmert**
University of Copenhagen
zimmert@di.ku.dk

**Yevgeny Seldin**
University of Copenhagen
seldin@di.ku.dk

## Abstract

We introduce the factored bandits model, which is a framework for learning with limited (bandit) feedback, where actions can be decomposed into a Cartesian product of atomic actions. Factored bandits incorporate rank-1 bandits as a special case, but significantly relax the assumptions on the form of the reward function. We provide an anytime algorithm for stochastic factored bandits and up to constants matching upper and lower regret bounds for the problem. Furthermore, we show how a slight modification enables the proposed algorithm to be applied to utility-based dueling bandits. We obtain an improvement in the additive terms of the regret bound compared to state-of-the-art algorithms (the additive terms are dominating up to time horizons that are exponential in the number of arms).

## 1 Introduction

We introduce *factored bandits*, which is a bandit learning model, where actions can be decomposed into a Cartesian product of atomic actions. As an example, consider an advertising task, where the actions can be decomposed into (1) selection of an advertisement from a pool of advertisements and (2) selection of a location on a web page out of a set of locations, where it can be presented. The probability of a click is then a function of the quality of the two actions, the attractiveness of the advertisement and the visibility of the location it was placed at. In order to maximize the reward the learner has to maximize the quality of actions along each dimension of the problem. Factored bandits generalize the above example to an arbitrary number of atomic actions and arbitrary reward functions satisfying some mild assumptions.

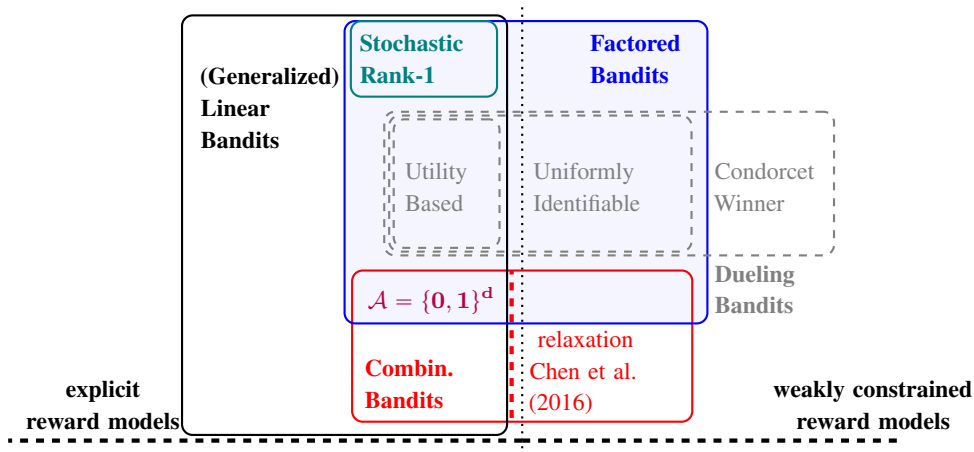

Figure 1: Relations between factored bandits and other bandit models.

In a nutshell, at every round of a factored bandit game the player selects $L$ atomic actions, $a_1, \ldots, a_L$, each from a corresponding finite set $\mathcal{A}_\ell$ of size $|\mathcal{A}_\ell|$ of possible actions. The player then observes a reward, which is an arbitrary function of $a_1, \ldots, a_L$ satisfying some mild assumptions. For example, it can be a sum of the quality of atomic actions, a product of the qualities, or something else that does not necessarily need to have an analytical expression. The learner does not have to know the form of the reward function.

Our way of dealing with combinatorial complexity of the problem is through introduction of *unique identifiability* assumption, by which the best action along each dimension is uniquely identifiable. A bit more precisely, when looking at a given dimension we call the collection of actions along all other dimensions a *reference set*. The unique identifiability assumption states that in expectation the best action along a dimension outperforms any other action along the same dimension by a certain margin when both are played with the same reference set, irrespective of the composition of the reference set. This assumption is satisfied, for example, by the reward structure in linear and generalized linear bandits, but it is much weaker than the linearity assumption.

In Figure 1, we sketch the relations between factored bandits and other bandit models. We distinguish between bandits with explicit reward models, such as linear and generalized linear bandits, and bandits with weakly constrained reward models, including factored bandits and some relaxations of combinatorial bandits. A special case of factored bandits are rank-1 bandits [7]. In rank-1 bandits the player selects two actions and the reward is the product of their qualities. Factored bandits generalize this to an arbitrary number of actions and significantly relax the assumption on the form of the reward function.

The relation with other bandit models is a bit more involved. There is an overlap between factored bandits and (generalized) linear bandits [1; 6], but neither is a special case of the other. When actions are represented by unit vectors, then for (generalized) linear reward functions the models coincide. However, the (generalized) linear bandits allow a continuum of actions, whereas factored bandits relax the (generalized) linearity assumption on the reward structure to uniform identifiability.

There is a partial overlap between factored bandits and combinatorial bandits [3]. The action set in combinatorial bandits is a subset of $\{0, 1\}^d$. If the action set is unrestricted, i.e. $\mathcal{A} = \{0, 1\}^d$, then combinatorial bandits can be seen as factored bandits with just two actions along each of the $d$ dimensions. However, typically in combinatorial bandits the action set is a strict subset of $\{0, 1\}^d$ and one of the parameters of interest is the permitted number of non-zero elements. This setting is not covered by factored bandits. While in the classical combinatorial bandits setting the reward structure is linear, there exist relaxations of the model, e.g. Chen et al. [4].

Dueling bandits are not directly related to factored bandits and, therefore, we depict them with faded dashed blocks in Figure 1. While the action set in dueling bandits can be decomposed into a product of the basic action set with itself (one for the first and one for the second action in the duel), the observations in dueling bandits are the identities of the winners rather than rewards. Nevertheless, we show that the proposed algorithm for factored bandits can be applied to utility-based dueling bandits.

The main contributions of the paper can be summarized as follows:

1. We introduce factored bandits and the uniform identifiability assumption.

2. Factored bandits with uniformly identifiable actions are a generalization of rank-1 bandits.

3. We provide an anytime algorithm for playing factored bandits under uniform identifiability assumption in stochastic environments and analyze its regret. We also provide a lower bound matching up to constants.

4. Unlike the majority of bandit models, our approach does not require explicit specification or knowledge of the form of the reward function (as long as the uniform identifiability assumption is satisfied). For example, it can be a weighted sum of the qualities of atomic actions (as in linear bandits), a product thereof, or any other function not necessarily known to the algorithm.

5. We show that the algorithm can also be applied to utility-based dueling bandits, where the additive factor in the regret bound is reduced by a multiplicative factor of $K$ compared to state-of-the-art (where $K$ is the number of actions). It should be emphasized that in state-of-the-art regret bounds for utility-based dueling bandits the additive factor is dominating

for time horizons below $\Omega(\exp(K))$, whereas in the new result it is only dominant for time horizons up to $\mathcal{O}(K)$.

6. Our work provides a unified treatment of two distinct bandit models: rank-1 bandits and utility-based dueling bandits.

The paper is organized in the following way. In Section 2 we introduce the factored bandit model and uniform identifiability assumption. In Section 3 we provide algorithms for factored bandits and dueling bandits. In Section 4 we analyze the regret of our algorithm and provide matching upper and lower regret bounds. In Section 5 we compare our work empirically and theoretically with prior work. We finish with a discussion in Section 6.

## 2 Problem Setting

### 2.1 Factored bandits

We define the game in the following way. We assume that the set of actions $\mathcal{A}$ can be represented as a Cartesian product of atomic actions, $\mathcal{A} = \bigotimes_{\ell=1}^{L} \mathcal{A}^{\ell}$. We call the elements of $\mathcal{A}^{\ell}$ *atomic arms*. For rounds $t = 1, 2, \ldots$ the player chooses an action $\mathbf{A}_t \in \mathcal{A}$ and observes a reward $r_t$ drawn according to an unknown probability distribution $p_{\mathbf{A}_t}$ (i.e., the game is "stochastic"). We assume that the mean rewards $\mu(\mathbf{a}) = \mathbb{E}[r_t | \mathbf{A}_t = \mathbf{a}]$ are bounded in $[-1, 1]$ and that the noise $\eta_t = r_t - \mu(\mathbf{A}_t)$ is conditionally 1-sub-Gaussian. Formally, this means that

$$\forall \lambda \in \mathbb{R} \qquad \mathbb{E}\left[e^{\lambda \eta_t} | \mathcal{F}_{t-1}\right] \leqslant \exp\left(\frac{\lambda^2}{2}\right),$$

where $\mathcal{F}_t := \{\mathbf{A}_1, r_1, \mathbf{A}_2, r_2, \ldots, \mathbf{A}_t, r_t\}$ is the filtration defined by the history of the game up to and including round $t$. We denote $\mathbf{a}^* = (a_1^*, a_2^*, \ldots, a_L^*) = \operatorname{argmax}_{\mathbf{a} \in \mathcal{A}} \mu(\mathbf{a})$.

**Definition 1** (uniform identifiability). *An atomic set $\mathcal{A}^k$ has a uniformly identifiable best arm $a_k^*$ if and only if*

$$\forall a \in \mathcal{A}^k \backslash \{a_k^*\} : \Delta_k(a) := \min_{\mathbf{b} \in \bigotimes_{\ell \neq k} \mathcal{A}^{\ell}} \mu(a_k^*, \mathbf{b}) - \mu(a, \mathbf{b}) > 0. \tag{1}$$

We assume that all atomic sets have uniformly identifiable best arms. The goal is to minimize the pseudo-regret, which is defined as

$$\operatorname{Reg}_T = \mathbb{E}\left[\sum_{t=1}^{T} \mu(\mathbf{a}^*) - \mu(\mathbf{A}_t)\right].$$

Due to generality of the uniform identifiability assumption we cannot upper bound the instantaneous regret $\mu(\mathbf{a}^*) - \mu(\mathbf{A}_t)$ in terms of the gaps $\Delta_{\ell}(a_{\ell})$. However, a sequential application of (1) provides a lower bound

$$\mu(\mathbf{a}^*) - \mu(\mathbf{a}) = \mu(\mathbf{a}^*) - \mu(a_1, a_2^*, \ldots, a_L^*) + \mu(a_1, a_2^*, \ldots, a_L^*) - \mu(\mathbf{a})$$

$$\geqslant \Delta_1(a_1) + \mu(a_1, a_2^*, \ldots, a_L^*) - \mu(\mathbf{a}) \geqslant \ldots \geqslant \sum_{\ell=1}^{L} \Delta_{\ell}(a_{\ell}). \tag{2}$$

For the upper bound let $\kappa$ be a problem dependent constant, such that $\mu(\mathbf{a}^*) - \mu(\mathbf{a}) \leqslant \kappa \sum_{\ell=1}^{L} \Delta_{\ell}(a_{\ell})$ holds for all $\mathbf{a}$. Since the mean rewards are in $[-1, 1]$, the condition is always satisfied by $\kappa = \min_{\mathbf{a}, \ell} 2\Delta_{\ell}^{-1}(a_{\ell})$ and by equation (2) $\kappa$ is always larger than 1. The constant $\kappa$ appears in the regret bounds. In the extreme case when $\kappa = \min_{\mathbf{a}, \ell} 2\Delta_{\ell}^{-1}(a_{\ell})$ the regret guarantees are fairly weak. However, in many specific cases mentioned in the previous section, $\kappa$ is typically small or even 1. We emphasize that algorithms proposed in the paper do not require the knowledge of $\kappa$. Thus, the dependence of the regret bounds on $\kappa$ is not a limitation and the algorithms automatically adapt to more favorable environments.

## 2.2 Dueling bandits

The set of actions in dueling bandits is factored into $\mathcal{A} \times \mathcal{A}$. However, strictly speaking the problem is not a factored bandit problem, because the observations in dueling bandits are not the rewards.[1] When playing two arms, $a$ and $b$, we observe the identity of the winning arm, but the regret is typically defined via average relative quality of $a$ and $b$ with respect to a "best" arm in $\mathcal{A}$.

The literature distinguishes between different dueling bandit settings. We focus on *utility-based dueling bandits* [14] and show that they satisfy the uniform identifiability assumption.

In utility-based dueling bandits, it is assumed that each arm has a utility $u(a)$ and that the winning probabilities are defined by $\mathbb{P}[a \text{ wins against } b] = \nu(u(a) - u(b))$ for a monotonously increasing link function $\nu$. Let $w(a, b)$ be 1 if $a$ wins against $b$ and 0 if $b$ wins against $a$. Let $a^* := \operatorname{argmax}_{a \in \mathcal{A}} u(a)$ denote the best arm. Then for any arm $b \in \mathcal{A}$ and any $a \in \mathcal{A} \backslash a^*$, it holds that $\mathbb{E}[w(a^*, b)] - \mathbb{E}[w(a, b)] = \nu(u(a^*) - u(b)) - \nu(u(a) - u(b)) > 0$, which satisfies the uniform identifiability assumption. For the rest of the paper we consider the linear link function $\nu(x) = \frac{1+x}{2}$. The regret is then defined by

$$\text{Reg}_T = \mathbb{E} \left[ \sum_{t=1}^{T} \frac{u(a^*) - u(A_t)}{2} + \frac{u(a^*) - u(B_t)}{2} \right]. \tag{3}$$

## 3 Algorithms

Although in theory an asymptotically optimal algorithm for any structured bandit problem was presented in [5], for factored bandits this algorithm does not only require solving an intractable semi-infinite linear program at every round, but it also suffers from additive constants which are exponential in the number of atomic actions $L$. An alternative naive approach could be an adaptation of sparring [16], where each factor runs an independent $K$-armed bandit algorithm and does not observe the atomic arm choices of other factors. The downside of sparring algorithms, both theoretically and practically, is that each algorithm operates under limited information and the rewards become non i.i.d. from the perspective of each individual factor.

Our Temporary Elimination Algorithm (TEA, Algorithm 1) avoids these downsides. It runs independent instances of the Temporary Elimination Module (TEM, Algorithm 3) in parallel, one per each factor of the problem. Each TEM operates on a single atomic set. The TEA is responsible for the synchronization of TEM instances. Two main ingredients ensure information efficiency. First, we use relative comparisons between arms instead of comparing absolute mean rewards. This cancels out the effect of non-stationary means. The second idea is to use local randomization in order to obtain unbiased estimates of the relative performance without having to actually play each atomic arm with the same reference, which would have led to prohibitive time complexity.

```
1  ∀ℓ : TEMℓ ← new TEM(𝒜ℓ)
2  t ← 1
3  for s = 1, 2, . . . do
4  │   Ms ←
   │     argmaxℓ | TEMℓ . getActiveSet(f(t)⁻¹)|
5  │   Ts ← (t, t + 1, . . . , t + Ms − 1)
6  │   for ℓ ∈ {1, . . . , L} in parallel do
7  │   │   TEMℓ . scheduleNext(Ts)
8  │   for t ∈ Ts do
9  │   │   rt ← play((TEMℓ .At)ℓ=1,...,L)
10 │   for ℓ ∈ {1, . . . , L} in parallel do
11 │   │   TEMℓ . feedback((rt′)t′∈Ts)
12 │   t ← t + |Ts|
```
**Algorithm 1:** Factored Bandit TEA

```
1  TEM ← new TEM(𝒜)
2  t ← 1
3  for s = 1, 2, . . . do
4  │   𝒜s ← TEM . getActiveSet(f(t)⁻¹)
5  │   Ts ← (t, t + 1, . . . , t + |𝒜s| − 1)
6  │   TEM . scheduleNext(Ts)
7  │   for b ∈ 𝒜s do
8  │   │   rt ← play(TEM .At, b)
9  │   │   t ← t + 1
10 │   TEM . feedback((rt′)t′∈Ts)
```
**Algorithm 2:** Dueling Bandit TEA

The TEM instances run in parallel in externally synchronized phases. Each module selects active arms in *getActiveSet*($\delta$), such that the optimal arm is included with high probability. The length of a phase is chosen such that each module can play each potentially optimal arm at least once in every phase. All modules schedule all arms for the phase in *scheduleNext*. This is done by choosing arms in a round robin fashion (random choices if not all arms can be played equally often) and ordering them randomly. All scheduled plays are executed and the modules update their statistics through the call of *feedback* routine. The modules use slowly increasing lower confidence bounds for the gaps in order to temporarily eliminate arms that are with high probability suboptimal. In all algorithms, we use $f(t) := (t+1)\log^2(t+1)$.

**Dueling bandits**   For dueling bandits we only use a single instance of TEM. In each phase the algorithm generates two random permutations of the active set and plays the corresponding actions from the two lists against each other. (The first permutation is generated in Line 6 and the second in Line 7 of Algorithm 2.)

## 3.1   TEM

The TEM tracks empirical differences between rewards of all arms $a_i$ and $a_j$ in $D_{ij}$. Based on these differences, it computes lower confidence bounds for all gaps. The set $\mathcal{K}^*$ contains those arms where all LCB gaps are zero. Additionally the algorithm keeps track of arms that were never removed from $\mathcal{B}$. During a phase, each arm from $\mathcal{K}^*$ is played at least once, but only arms in $\mathcal{B}$ can be played more than once. This is necessary to keep the additive constants at $M\log(K)$ instead of $MK$.

**global** : $N_{i,j}, D_{i,j}, \mathcal{K}^*, \mathcal{B}$

1 **Function** initialize($\mathcal{K}$)
2     $\forall a_i, a_j \in \mathcal{K} : N_{i,j}, D_{i,j} \leftarrow 0, 0$
3     $\mathcal{B} \leftarrow \mathcal{K}$
4
5 **Function** getActiveSet($\delta$)
6     **if** $\exists N_{i,j} = 0$ **then**
7        $\mathcal{K}^* \leftarrow \mathcal{K}$
8     **else**
9        **for** $a_i \in \mathcal{K}$ **do**
10           $\hat{\Delta}^{LCB}(a_i) \leftarrow \max_{a_j \neq a_i} \frac{D_{j,i}}{N_{j,i}} - \sqrt{\frac{12\log(2Kf(N_{j,i})\delta^{-1})}{N_{j,i}}}$
11        $\mathcal{K}^* \leftarrow \{a_i \in \mathcal{K} | \hat{\Delta}^{LCB}(a_i) \leqslant 0\}$
12        **if** $|\mathcal{K}^*| = 0$ **then**
13           $\mathcal{K}^* \leftarrow \mathcal{K}$
14        $\mathcal{B} \leftarrow \mathcal{B} \cap \mathcal{K}^*$
15        **if** $|\mathcal{B}| = 0$ **then**
16           $\mathcal{B} \leftarrow \mathcal{K}^*$
17     **return** $\mathcal{K}^*$
18

19 **Function** scheduleNext($\mathcal{T}$)
20     **for** $a \in \mathcal{K}^*$ **do**
21        $\tilde{t} \leftarrow$ random unassigned index in $\mathcal{T}$
22        $A_{\tilde{t}} \leftarrow a$
23     **while** *not all* $A_{t_s}, \ldots, A_{t_s+|\mathcal{T}|-1}$ *assigned* **do**
24        **for** $a \in \mathcal{B}$ **do**
25           $\tilde{t} \leftarrow$ random unassigned index in $\mathcal{T}$
26           $A_{\tilde{t}} \leftarrow a$
27
28 **Function** feedback($\{R_t\}_{t_s,\ldots,t_s+M_s-1}$)
29     $\forall a_i : N_s^i, R_s^i \leftarrow 0, 0$
30     **for** $t = t_s, \ldots, t_s + M_s - 1$ **do**
31        $R_s^{A_t} \leftarrow R_s^{A_t} + R_t$
32        $N_s^{A_t} \leftarrow N_s^{A_t} + 1$
33     **for** $a_i, a_j \in \mathcal{K}^*$ **do**
34        $D_{i,j} \leftarrow D_{i,j} + \min\{N_i^s, N_j^s\}(\frac{R_s^i}{N_s^i} - \frac{R_s^j}{N_s^j})$
35        $N_{i,j} \leftarrow N_{i,j} + \min\{N_i^s, N_j^s\}$

**Algorithm 3:** Temporary Elimination Module (TEM) Implementation

# 4   Analysis

We start this section with the main theorem, which bounds the number of times the TEM pulls sub-optimal arms. Then we prove upper bounds on the regret for our main algorithms. Finally, we prove a lower bound for factored bandits that shows that our regret bound is tight up to constants.

## 4.1   Upper bound for the number of sub-optimal pulls by TEM

**Theorem 1.** *For any TEM submodule* $\text{TEM}^\ell$ *with an arm set of size* $K = |\mathcal{A}^\ell|$, *running in the TEA algorithm with* $M := \max_\ell |\mathcal{A}^\ell|$ *and any suboptimal atomic arm* $a \neq a^*$, *let* $N_t(a)$ *denote the number of times TEM has played the arm* $a$ *up to time* $t$. *Then there exist constants* $C(a) \leqslant M$ *for*

$a \neq a^*$, *such that*

$$\mathbb{E}[N_t(a)] \leqslant \frac{120}{\Delta(a)^2} \left( \log(2Kt\log^2(t)) + 4\log \left( \frac{48\log(2Kt\log^2(t))}{\Delta(a)^2} \right) \right) + C(a),$$

*where* $\sum_{a \neq a^*} C(a) \leqslant M\log(K) + \frac{5}{2}K$ *in the case of factored bandits and* $C(a) \leqslant \frac{5}{2}$ *for dueling bandits.*

*Proof sketch.* [The complete proof is provided in the Appendix.]

**Step 1**  We show that the confidence intervals are constructed in such a way that the probability of all confidence intervals holding at all epochs up from $s'$ is at least $1 - \max_{s \geqslant s'} f(t_s)^{-1}$. This requires a novel concentration inequality (Lemma 3) for a sum of conditionally $\sigma_s$-sub-gaussian random variables, where $\sigma_s$ can be dependent on the history. This technique might be useful for other problems as well.

**Step 2**  We split the number of pulls into pulls that happen in rounds where the confidence intervals hold and those where they fail: $N_t(a) = N_t^{conf}(a) + N_t^{\overline{conf}}(a)$.

We can bound the expectation of $N_t^{\overline{conf}}(a)$ based on the failure probabilities given by $\mathbb{P}[\text{conf failure at round s}] \leqslant \frac{1}{f(t_s)}$.

**Step 3**  We define $s'$ as the last round in which the confidence intervals held and $a$ was not eliminated. We can split $N_t^{conf}(a) = N_{t_{s'}}^{conf}(a) + C(a)$ and use the confidence intervals to upper bound $N_{t_{s'}}^{conf}(a)$. The upper bound on $\sum_a C(a)$ requires special handling of arms that were eliminated once and carefully separating the cases where confidence intervals never fail and those where they might fail. □

## 4.2  Regret Upper bound for Dueling Bandit TEA

A regret bound for the Factored Bandit TEA algorithm, Algorithm 1, is provided in the following theorem.

**Theorem 2.** *The pseudo-regret of Algorithm 1 at any time $T$ is bounded by*

$$\mathrm{Reg}_T \leqslant \kappa \left( \sum_{\ell=1}^{L} \sum_{a_\ell \neq a_\ell^*} \frac{120}{\Delta_\ell(a_\ell)} \left( \log(2|\mathcal{A}^\ell|t\log^2(t)) + 4\log \left( \frac{48\log(2|\mathcal{A}^\ell|t\log^2(t))}{\Delta_\ell(a_\ell)} \right) \right) \right)$$

$$+ \max_\ell |\mathcal{A}^\ell| \sum_\ell \log(|\mathcal{A}^\ell|) + \sum_\ell \frac{5}{2}|\mathcal{A}^\ell|.$$

*Proof.* The design of TEA allows application of Theorem 1 to each instance of TEM. Using $\mu(\mathbf{a}_*) - \mu(\mathbf{a}) \leqslant \kappa \sum_{\ell=1}^{L} \Delta_\ell(a_\ell)$, we have that

$$\mathrm{Reg}_T = \mathbb{E}[\sum_{t=1}^{T} \mu(\mathbf{a}^*) - \mu(\mathbf{a}_t)]] \leqslant \kappa \sum_{l=1}^{L} \sum_{a_\ell \neq a_\ell^*} \mathbb{E}[N_T(a_\ell)]\Delta_\ell(a_\ell).$$

Applying Theorem 1 to the expected number of pulls and bounding the sums $\sum_a C(a)\Delta(a) \leqslant \sum_a C(a)$ completes the proof. □

## 4.3  Dueling bandits

A regret bound for the Dueling Bandit TEA algorithm (DBTEA), Algorithm 2, is provided in the following theorem.

**Theorem 3.** *The pseudo-regret of Algorithm 2 for any utility-based dueling bandit problem at any time $T$ (defined in equation (3) satisfies* $\mathrm{Reg}_T \leqslant \mathcal{O} \left( \sum_{a \neq a^*} \frac{\log(T)}{\Delta(a)} \right) + \mathcal{O}(K)$.

*Proof.* At every round, each arm in the active set is played once in position $A$ and once in position $B$ in $play(A, B)$. Denote by $N_t^A(a)$ the number of plays of an arm $a$ in the first position, $N_t^B(a)$ the number of plays in the second position, and $N_t(a)$ the total number of plays of the arm. We have

$$\text{Reg}_T = \sum_{a \neq a_*} \mathbb{E}[N_t(a)]\Delta(a) = \sum_{a \neq a_*} \mathbb{E}[N_t^A(a) + N_t^B(a)]\Delta(a) = \sum_{a \neq a_*} 2\mathbb{E}[N_t^A(a)]\Delta(a).$$

The proof is completed by applying Theorem 1 to bound $\mathbb{E}[N_t^A(a)]$. $\qquad\square$

## 4.4 Lower bound

We show that without additional assumptions the regret bound cannot be improved. The lower bound is based on the following construction. The mean reward of every arm is given by $\mu(\mathbf{a}) = \mu(\mathbf{a^*}) - \sum_\ell \Delta_\ell(a_\ell)$. The noise is Gaussian with variance 1. In this problem, the regret can be decomposed into a sum over atomic arms of the regret induced by pulling these arms, $\text{Reg}_T = \sum_\ell \sum_{a_\ell \in \mathcal{A}^\ell} \mathbb{E}[N_T(a_\ell)]\Delta_\ell(a_\ell)$. Assume that we only want to minimize the regret induced by a single atomic set $\mathcal{A}^\ell$. Further, assume that $\Delta_k(a)$ for all $k \neq \ell$ are given. Then the problem is reduced to a regular $K$-armed bandit problem. The asymptotic lower bound for $K$-armed bandit under 1-Gaussian noise goes back to [10]: For any consistent strategy $\theta$, the asymptotic regret is lower bounded by $\liminf_{T \to \infty} \frac{\text{Reg}_T^\theta}{\log(T)} \geqslant \sum_{a \neq a_*} \frac{2}{\Delta(a)}$. Due to regret decomposition, we can apply this bound to every atomic set separately. Therefore, the asymptotic regret in the factored bandit problem is

$$\liminf_{T \to \infty} \frac{\text{Reg}_T^\theta}{\log(T)} \geqslant \sum_{\ell=1}^{L} \sum_{a^\ell \neq a_*^\ell} \frac{2}{\Delta^\ell(a^\ell)}.$$

This shows that our general upper bound is asymptotically tight up to leading constants and $\kappa$.

$\kappa$**-gap**    We note that there is a problem-dependent gap of $\kappa$ between our upper and lower bounds. Currently we believe that this gap stems from the difference between information and computational complexity of the problem. Our algorithm operates on each factor of the problem independently of other factors and is based on the "optimism in the face of uncertainty" principle. It is possible to construct examples in which the optimal strategy requires playing surely sub-optimal arms for the sake of information gain. For example, this kind of constructions were used by Lattimore and Szepesvári [11] to show suboptimality of optimism-based algorithms. Therefore, we believe that removing $\kappa$ from the upper bound is possible, but requires a fundamentally different algorithm design. What is not clear is whether it is possible to remove $\kappa$ without significant sacrifice of the computational complexity.

## 5    Comparison to Prior Work

### 5.1    Stochastic rank-1 bandits

Stochastic rank-1 bandits introduced by Katariya et al. [7] are a special case of factored bandits. The authors published a refined algorithm for Bernoulli rank-1 bandits using KL confidence sets in Katariya et al. [8]. We compare our theoretical results with the first paper because it matches our problem assumptions. In our experiments, we provide a comparison to both the original algorithm and the KL version.

In the stochastic rank-1 problem there are only 2 atomic sets of size $K_1$ and $K_2$. The matrix of expected rewards for each pair of arms is of rank 1. It means that for each $u \in \mathcal{A}^1$ and $v \in \mathcal{A}^2$, there exist $\overline{u}, \overline{v} \in [0, 1]$ such that $\mathbb{E}[r(u, v)] = \overline{u} \cdot \overline{v}$. The proposed Stochastic rank-1 Elimination algorithm introduced by Katariya et al. is a typical elimination style algorithm. It requires knowledge of the time horizon and uses phases that increase exponentially in length. In each phase, all arms are played uniformly. At the end of a phase, all arms that are sub-optimal with high probability are eliminated.

**Theoretical comparison**    It is hard to make a fair comparison of the theoretical bounds because TEA operates under much weaker assumptions. Both algorithms have a regret bound of $\mathcal{O}\left(\left(\sum_{u \in \mathcal{A}^1 \setminus u*} \frac{1}{\Delta_1(u)} + \sum_{v \in \mathcal{A}^2 \setminus v*} \frac{1}{\Delta_2(v)}\right) \log(t)\right)$. The problem independent multiplicative factors

hidden under $\mathcal{O}$ are smaller for TEA, even without considering that rank-1 Elimination requires a doubling trick for anytime applications. However, the problem dependent factors are in favor of rank-1 Elimination, where the gaps correspond to the mean difference under uniform sampling $(\overline{u}^* - \overline{u}) \sum_{v \in \mathcal{A}^2} \overline{v}/K_2$. In factored bandits, the gaps are defined as $(\overline{u}^* - \overline{u}) \min_{v \in \mathcal{A}^2} \overline{v}$, which is naturally smaller. The difference stems from different problem assumptions. Stronger assumptions of rank-1 bandits make elimination easier as the number of eliminated suboptimal arms increases. The TEA analysis holds in cases where it becomes harder to identify suboptimal arms after removal of bad arms. This may happen when highly suboptimal atomic actions in one factor provide more discriminative information on atomic actions in other factors than close to optimal atomic actions in the same factor (this follows the spirit of illustration of suboptimality of optimistic algorithms in [11]). We leave it to future work to improve the upper bound of TEA under stronger model assumptions.

In terms of memory and computational complexity, TEA is inferior to regular elimination style algorithms, because we need to keep track of relative performances of the arms. That means both computational and memory complexities are $\mathcal{O}(\sum_\ell |\mathcal{A}^\ell|^2)$ per round in the worst case, as opposed to rank-1 Elimination that only requires $\mathcal{O}\left(|\mathcal{A}^1| + |\mathcal{A}^2|\right)$.

**Empirical comparison** The number of arms is set to 16 in both sets. We always fix $\overline{u}^* - \overline{u} = \overline{v}^* - \overline{v} = 0.2$. We vary the absolute value of $\overline{u}^*\overline{v}^*$. As expected, rank1ElimKL has an advantage when the Bernoulli random variables are strongly biased towards one side. When the bias is close to $\frac{1}{2}$, we clearly see the better constants of TEA. In the evaluation we clearly outperform rank-1 Elimination

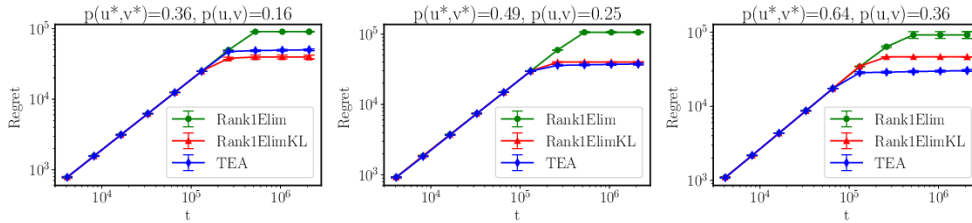

Figure 2: Comparison of Rank1Elim, Rank1ElimKL, and TEA for $K = L = 16$. The results are averaged over 20 repetitions of the experiment.

over different parameter settings and even beat the KL optimized version if the means are not too close to zero or one. This supports that our algorithm does not only provide a more practical anytime version of elimination, but also improves on constant factors in the regret. We believe that our algorithm design can be used to improve other elimination style algorithms as well.

## 5.2 Dueling Bandits: Related Work

To the best of our knowledge, the proposed Dueling Bandit TEA is the first algorithm that satisfies the following three criteria simultaneously for utility-based dueling bandits:

- It requires no prior knowledge of the time horizon (nor uses the doubling trick or restarts).
- Its pseudo-regret is bounded by $\mathcal{O}(\sum_{a \neq a*} \frac{\log(t)}{\Delta(a)})$.
- There are no additive constants that dominate the regret for time horizons $T > \mathcal{O}(K)$.

We want to stress the importance of the last point. For all state-of-the-art algorithms known to us, when the number of actions $K$ is moderately large, the additive term is dominating for any realistic time horizon $T$. In particular, Ailon et al. [2] introduces three algorithms for the utility-based dueling bandit problem. The regret of Doubler scales with $\mathcal{O}(\log^2(t))$. The regret of MultiSBM has an additive term of order $\sum_{a \neq a*} \frac{K}{\Delta(a)}$ that is dominating for $T < \Omega(\exp(K))$. The last algorithm, Sparring, has no theoretical analysis.

Algorithms based on the weaker Condorcet winner assumption apply to utility-based setting, but they all suffer from equally large or even larger additive terms. The RUCB algorithm introduced by Zoghi et al. [17] has an additive term in the bound that is defined as $2D\Delta_{max} \log(2D)$, for

$\Delta_{max} = \max_{a \neq a*} \Delta(a)$ and $D > \frac{1}{2} \sum_{a_i \neq a*} \sum_{a_j \neq a_i} \frac{4\alpha}{\min\{\Delta(a_i)^2, \Delta(a_j)^2\}}$. By unwrapping these definitions, we see that the RUCB regret bound has an additive term of order $2D\Delta_{max} \geqslant \sum_{a \neq a*} \frac{K}{\Delta(a)}$. This is again the dominating term for time horizons $T \leqslant \Omega(\exp(K))$. The same applies to the RMED algorithm introduced by Komiyama et al. [9], which has an additive term of $\mathcal{O}(K^2)$. (The dependencies on the gaps are hidden behind the $\mathcal{O}$-notation.) The D-TS algorithm by Wu and Liu [13] based on Thompson Sampling shows one of the best empirical performances, but its regret bound includes an additive constant of order $\mathcal{O}(K^3)$.

Other algorithms known to us, Interleaved Filter [16], Beat the Mean [15], and SAVAGE [12], all require knowledge of the time horizon $T$ in advance.

**Empirical comparison**   We have used the framework provided by Komiyama et al. [9]. We use the same utility for all sub-optimal arms. In Figure 3, the winning probability of the optimal arm over suboptimal arms is always set to $0.7$, we run the experiment for different number of arms $K$. TEA outperforms all algorithms besides RMED variants, as long as the number of arms are sufficiently big. To show that there also exists a regime where the improved constants gain an advantage over RMED, we conducted a second experiment in Figure 4 (in the Appendix), where we set the winning probability to $0.95^2$ and significantly increase the number of arms. The evaluation shows that the additive terms are indeed non-negligible and that Dueling Bandit TEA outperforms all baseline algorithms when the number of arms is sufficiently large.

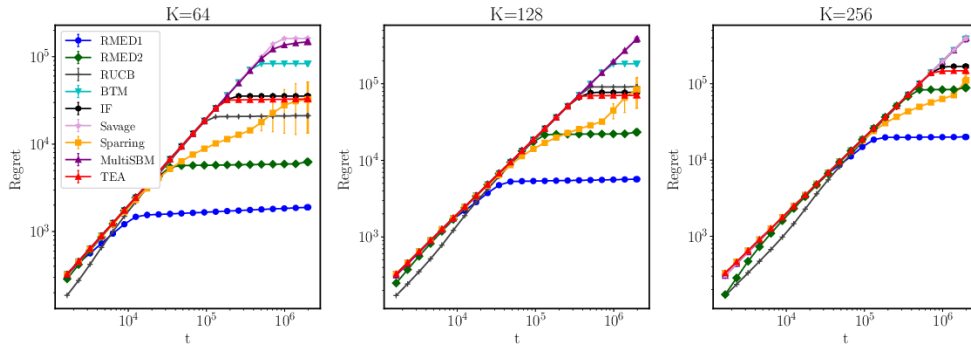

Figure 3: Comparison of Dueling Bandits algorithms with identical gaps of $0.4$. The results are averaged over 20 repetitions of the experiment.

# 6   Discussion

We have presented the factored bandits model and uniform identifiability assumption, which requires no knowledge of the reward model. We presented an algorithm for playing stochastic factored bandits with uniformly identifiable actions and provided matching upper and lower bounds for the problem up to constant factors. Our algorithm and proofs might serve as a template to turn other elimination style algorithms into improved anytime algorithms.

Factored bandits with uniformly identifiable actions generalize rank-1 bandits. We have also provided a unified framework for the analysis of factored bandits and utility-based dueling bandits. Furthermore, we improve the additive constants in the regret bound compared to state-of-the-art algorithms for utility-based dueling bandits.

There are multiple potential directions for future research. One example mentioned in the text is the possibility of improving the regret bound when additional restrictions on the form of the reward function are introduced or improvements of the lower bound when algorithms are restricted in computational or memory complexity. Another example is the adversarial version of the problem.

## Footnotes

[1]In principle, it is possible to formulate a more general problem that would incorporate both factored bandits and dueling bandits. But such a definition becomes too general and hard to work with. For the sake of clarity we have avoided this path.

[2]Smaller gaps show the same behavior but require more arms and more timesteps.

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
