[Supplementary Material]

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

4      Mₛ ←
          argmaxₗ |TEMˡ.getActiveSet(f(t)⁻¹)|
5      Tₛ ← (t, t+1, …, t + Mₛ − 1)
6      for ℓ ∈ {1, …, L} in parallel do
7          TEMˡ.scheduleNext(Tₛ)
8      for t ∈ Tₛ do
9          rₜ ← play((TEMˡ.Aₜ)ₗ₌₁,…,ₗ)
10     for ℓ ∈ {1, …, L} in parallel do
11         TEMˡ.feedback((rₜ')ₜ'∈Tₛ)
12     t ← t + |Tₛ|
        Algorithm 1: Factored Bandit TEA
```

```
1  TEM ← new TEM(𝒜)
2  t ← 1
3  for s = 1, 2, … do
4      𝒜ₛ ← TEM.getActiveSet(f(t)⁻¹)
5      Tₛ ← (t, t+1, …, t + |𝒜ₛ| − 1)
6      TEM.scheduleNext(Tₛ)
7      for b ∈ 𝒜ₛ do
8          rₜ ← play(TEM.Aₜ, b)
9          t ← t + 1
10     TEM.feedback((rₜ')ₜ'∈Tₛ)
        Algorithm 2: Dueling Bandit TEA
```

The TEM instances run in parallel in externally synchronized phases. Each module selects active arms in *getActiveSet($\delta$)*, such that the optimal arm is included with high probability. The length of a phase is chosen such that each module can play each potentially optimal arm at least once in every phase. All modules schedule all arms for the phase in *scheduleNext*. This is done by choosing arms in a round robin fashion (random choices if not all arms can be played equally often) and ordering them randomly. All scheduled plays are executed and the modules update their statistics through the call of *feedback* routine. The modules use slowly increasing lower confidence bounds for the gaps in order to temporarily eliminate arms that are with high probability suboptimal. In all algorithms, we use $f(t) := (t+1)\log^2(t+1)$.

**Dueling bandits**   For dueling bandits we only use a single instance of TEM. In each phase the algorithm generates two random permutations of the active set and plays the corresponding actions from the two lists against each other. (The first permutation is generated in Line 6 and the second in Line 7 of Algorithm 2.)

## 3.1   TEM

The TEM tracks empirical differences between rewards of all arms $a_i$ and $a_j$ in $D_{ij}$. Based on these differences, it computes lower confidence bounds for all gaps. The set $\mathcal{K}^*$ contains those arms where all LCB gaps are zero. Additionally the algorithm keeps track of arms that were never removed from $\mathcal{B}$. During a phase, each arm from $\mathcal{K}^*$ is played at least once, but only arms in $\mathcal{B}$ can be played more than once. This is necessary to keep the additive constants at $M\log(K)$ instead of $MK$.

**global** $: N_{i,j}, D_{i,j}, \mathcal{K}^*, \mathcal{B}$

1 **Function** `initialize`($\mathcal{K}$)
2 $\quad \forall a_i, a_j \in \mathcal{K} : N_{i,j}, D_{i,j} \leftarrow 0, 0$
3 $\quad \mathcal{B} \leftarrow \mathcal{K}$
4
5 **Function** `getActiveSet`($\delta$)
6 $\quad$ **if** $\exists N_{i,j} = 0$ **then**
7 $\quad\quad | \quad \mathcal{K}^* \leftarrow \mathcal{K}$
8 $\quad$ **else**
9 $\quad\quad$ **for** $a_i \in \mathcal{K}$ **do**
10 $\quad\quad\quad | \quad \hat{\Delta}^{LCB}(a_i) \leftarrow \max_{a_j \neq a_i} \frac{D_{j,i}}{N_{j,i}} - \sqrt{\frac{12\log(2Kf(N_{j,i})\delta^{-1})}{N_{j,i}}}$
11 $\quad\quad\quad \mathcal{K}^* \leftarrow \{a_i \in \mathcal{K} | \hat{\Delta}^{LCB}(a_i) \leqslant 0\}$
12 $\quad\quad\quad$ **if** $|\mathcal{K}^*| = 0$ **then**
13 $\quad\quad\quad\quad | \quad \mathcal{K}^* \leftarrow \mathcal{K}$
14 $\quad\quad\quad \mathcal{B} \leftarrow \mathcal{B} \cap \mathcal{K}^*$
15 $\quad\quad\quad$ **if** $|\mathcal{B}| = 0$ **then**
16 $\quad\quad\quad\quad | \quad \mathcal{B} \leftarrow \mathcal{K}^*$
17 $\quad$ **return** $\mathcal{K}^*$
18

19 **Function** `scheduleNext`($\mathcal{T}$)
20 $\quad$ **for** $a \in \mathcal{K}^*$ **do**

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

# A Auxiliary Lemmas

**Lemma 1.** *Given positive real numbers $\sigma_1, \sigma_2, \ldots, \sigma_n..$ If $(X_i)_{i=1,\ldots,n}$ is a sequence of random variables such that $X_i$ conditioned on $X_{i-1}, X_{i-2}, \ldots$ is $\sigma_i$-sub-Gaussian. Then $Z = \sum_{i=1}^{n} X_i$ is $\sqrt{\sum_{i=1}^{n} \sigma_i^2}$-sub-Gaussian.*

We believe this is a standard result, however we only found references for independent sub-Gaussian random variables.

*Proof of Lemma 1.* For $t = 1, \ldots, n$ define $M_{s,t} = \exp(s \sum_{i=1}^{t} X_i - \frac{1}{2} \sum_{i=1}^{t} s^2 \sigma_i^2)$. We claim $M_{s,t}$ is a super-martingale. Given that $X_i$ are conditionally sub-Gaussian, we have $\mathbb{E}[\exp(sX_{t+1})|X_t, X_{t-1}, \ldots] \leqslant \exp(\frac{s^2 \sigma_{t+1}^2}{2})$. So

$$\mathbb{E}[M_{s,t+1}|M_{s,t}] = \mathbb{E}[\exp(sX_{t+1} - \frac{1}{2}s^2 \sigma_{t+1}^2)M_{s,t}|M_{s,t}]$$

$$= \mathbb{E}[\exp(sX_{t+1} - \frac{1}{2}s^2 \sigma_{t+1}^2)|M_{s,t}]M_{s,t} \leqslant M_{s,t}.$$

Additionally by definition of sub-Gaussian $\mathbb{E}[M_{s,1}] \leqslant 1$. Therefore $\mathbb{E}[M_{s,n}] \leqslant 1$. Finally we get that $\mathbb{E}[\exp(sZ)] = \mathbb{E}[M_{s,n} \cdot \exp(\sum_{i=1}^{n} \frac{s^2 \sigma_i^2}{2})] \leqslant \exp(\sum_{i=1}^{n} \frac{s^2 \sigma_i^2}{2})$. So $Z$ is $\sqrt{\sum_{i=1}^{n} \sigma_i^2}$-sub-Gaussian. $\square$

**Lemma 2.** *Let $y \geqslant 1, z \geqslant 10$, then for any $x > zy + 4z \log(zy)$:*

$$\frac{z(\log(f(x)) + y)}{x} < 1.$$

*Proof.* We can reparameterize $x = zy + \alpha z \log(zy)$ for $\alpha > 4$. Then

$$\frac{zy + z \log(f(zy + \alpha z \log(zy)))}{zy + \alpha z \log(zy)} < 1$$

$$\Leftrightarrow \frac{\log(f(zy + \alpha z \log(zy)))}{\alpha \log(zy)} < 1$$

$$\Leftrightarrow f(zy + \alpha z \log(zy)) < (zy)^\alpha$$

$$\Leftarrow f(zy + \alpha zy \log(zy)) < (zy)^\alpha.$$

Using $\log(x) \leqslant \sqrt{x} - \frac{1}{2}$ and $\alpha > 4$, we have that

$$f(zy + \alpha zy \log(zy)) < f\left(zy + \alpha zy(\sqrt{zy} - \frac{1}{2})\right) < f(\alpha(zy)^{\frac{3}{2}} - 1) = \alpha(zy)^{\frac{3}{2}} \log^2(\alpha(zy)^{\frac{3}{2}}).$$

It is therefore sufficient to prove that for all $\tilde{x} > 10$ and $\alpha > 4$:

$$\alpha \log^2(\alpha \tilde{x}^{\frac{3}{2}}) < \tilde{x}^{\alpha - \frac{3}{2}}$$

$$\Leftarrow \alpha(\sqrt{\alpha} + \tilde{x}^{\frac{3}{4}})^2 < \tilde{x}^{\alpha - \frac{3}{2}}$$

$$\Leftrightarrow \sqrt{\alpha}(\sqrt{\alpha}\tilde{x}^{\frac{3}{4} - \frac{\alpha}{2}} + \tilde{x}^{\frac{3}{2} - \frac{\alpha}{2}}) < 1.$$

The minimum on the left hand side is obtained for $\alpha = 4$ and $\tilde{x} = 10$ for with it holds true. $\square$

**Lemma 3.** *Let $\sigma \in \mathbb{R}$ and $X_1, X_2, \ldots$ be a sequence of sub-Gaussian random variables adapted to the filtration $\mathcal{F}_1, \mathcal{F}_2, \ldots$, i.e. $\mathbb{E}[e^{sX_t}|X_1, X_2, \ldots, X_{t-1}] \leqslant e^{-\frac{\sigma_t^2 s^2}{2}}$. Assume for all $t$: $\sum_{i=1}^{t} \sigma_i^2 = n_t \sigma^2$, with $n_t \in \mathbb{N}$ almost surely. Then*

$$\mathbb{P}\left[\exists t \in \mathbb{N} : \sum_{i=1}^{t} X_i \geqslant \sqrt{2\sigma^2 n_t \log\left(\frac{f(n_t)}{\delta}\right)}\right] \leqslant \delta,$$

*where $f(n_t) = 2(1 + n_t) \log^2(1 + n_t)$.*

Note that unlike in Lemma 1, we do not require $\sigma_t$ to be independent of $X_1, \ldots, X_{t-1}$.

*Proof.* The proof follows closely the arguments presented in the proofs of Lemma 8 in Abbasi-Yadkori et al. [1] and Lemma 14 in Lattimore and Szepesvári [11]. For $\psi \in \mathbb{R}$ define

$$M_{t,\psi} = \exp\left(\sum_{s=1}^{t} \psi X_s - \frac{\psi^2 \sigma_s^2}{2}\right).$$

If $t_0 \leqslant \tau \leqslant t$ is a stopping time with respect to $\mathcal{F}$, then as in the proof of Abbasi-Yadkori et al. [1, Lemma 8] we have $\mathbb{E}[M_{\tau,\psi}] \leqslant 1$. By Markov's inequality, we have

$$\mathbb{P}[M_{\tau,\psi} \geqslant 1/\delta] \leqslant \delta \qquad \Leftrightarrow \qquad \mathbb{P}\left[\sum_{s=1}^{\tau} X_s \geqslant \frac{\log(\delta^{-1})}{\psi} + \frac{\psi n_\tau \sigma^2}{2}\right] \leqslant \delta.$$

An optimal choice of $\psi$ would be $\psi = \sqrt{2\frac{\log(1/\delta)}{n_\tau \sigma^2}}$, however $\psi X_t$ would not be $\mathcal{F}_t$-measurable for $t \leqslant \tau$ and $M_{t,\psi}$ would not be well defined. Instead, for $k \geqslant 1$ we define

$$\psi_k := \sqrt{\frac{2\log(f(k)\delta^{-1})}{k\sigma^2}}.$$

With a union bound, we get that

$$\mathbb{P}\left[\exists k \geqslant 1 : \sum_{s=1}^{\tau} X_s \geqslant \frac{\log(f(k)\delta^{-1})}{\psi_k} + \frac{\psi n_\tau \sigma^2}{2}\right] \leqslant \sum_{k=1}^{\infty} \frac{\delta}{f(k)} \leqslant \delta.$$

Using now $k = n_\tau$, for which this also holds, we get that

$$\mathbb{P}\left[\sum_{s=1}^{\tau} X_s \geqslant \sqrt{2\sigma^2 n_\tau \log\left(\frac{f(n_\tau)}{\delta}\right)}\right] \leqslant \delta.$$

The proof is completed by choosing a stopping time $\tau$:

$$\tau = \min\left(\infty \cup \left\{t \geqslant 1 : \sum_{s=1}^{t} X_s \geqslant \sqrt{2n_t \sigma^2 \log\left(\frac{f(n_t)}{\delta}\right)}\right\}\right).$$

$\square$

**Lemma 4.** *Given $X_1, X_2, \ldots, X_n$ random variables with means $p_1, p_2, \ldots, p_n \in [-1, 1]$, such that all $X_i - p_i$ are 1-sub-Gaussian. (e.g. Bernoulli random variables) Given further two sample sizes $m, k \geqslant 1$, such that $m + k \leqslant n$. Then for $I_m : |I_m| = m$ and $I_k : |I_k| = k$ disjoint uniform samples of indices in $(1, 2, \ldots, n)$ without replacement, the random variable*

$$Z = \frac{1}{m}\sum_{i \in I_m} X_i - \frac{1}{k}\sum_{i \in I_k} X_i,$$

*is $\sqrt{\frac{3(m+k)}{mk}}$-sub-Gaussian.*

*Proof.* Without loss of generality, we set $m \leqslant k$. By definition, the random variables $X_i$ can be decomposed into $X_i = p_i + \eta_i$, where $\eta_i$ are conditionally independent 1-sub-Gaussian random variables. Decomposing $Z$ gives:

$$Z = \frac{1}{m}\sum_{i \in I_m} p_i - \frac{1}{k}\sum_{i \in I_k} p_i + \frac{1}{m}\sum_{i \in I_m} \eta_i - \frac{1}{k}\sum_{i \in I_k} \eta_i.$$

We define $\bar{I} = \{1, ..., n\} \backslash (I_m \cup I_k)$, the indices of remaining $X_i$'s and $\bar{p} = \frac{1}{n}\sum_{i=1}^{n} p_i$ the mean of means. In order to show that $\frac{1}{m}\sum_{i \in I_m} p_i - \frac{1}{k}\sum_{i \in I_k} p_i$ is sub-Gaussian, we first draw the elements in

$(p_i)_{i \in \overline{I}} = (\overline{P}_i)_{i=1,\dots,n-m-k}$ and then the set $(p_i)_{i \in I_m} = (P_i^m)_{i=1,\dots,m}$. Drawing the first element $\overline{P}_1$ can be written as $\overline{P}_1 = \overline{p} + \zeta_1$, where $\zeta_1$ is sub-Gaussian. With continuous drawings, it holds that

$$\mathbb{E}[\overline{P}_2|\overline{P}_1] = \overline{p} - \frac{1}{n-1}\zeta_1$$

$$\overline{P}_2 = \overline{p} - \frac{1}{n-1}\zeta_1 + \zeta_2$$

$$\mathbb{E}[\overline{P}_3|\overline{P}_1,\overline{P}_2] = \overline{p} - \frac{1}{n-1}\zeta_1 - \frac{1}{n-2}\zeta_2$$

$$\overline{P}_3 = \overline{p} - \frac{1}{n-1}\zeta_1 - \frac{1}{n-2}\zeta_2 + \zeta_3$$

$$\dots$$

$$\mathbb{E}[\overline{P}_{n-m-k}|\overline{P}_1,\dots,\overline{P}_{n-m-k-1}] = \overline{p} - \sum_{i=1}^{n-m-k-1} \frac{1}{n-i}\zeta_i$$

$$\overline{P}_{n-m-k} = \overline{p} - \sum_{i=1}^{n-m-k-1} \frac{1}{n-i}\zeta_i + \zeta_{n-m-k}$$

$$\sum_{i=1}^{n-m-k} \overline{P}_i = (n-m-k)\overline{p} + \sum_{i=1}^{n-m-k} \frac{m+k}{n-i}\zeta_i$$

The noise variables $\zeta_i$ are all conditionally independent and 1-sub-Gaussian.

We continue with $P_i^m$ in the same fashion:

$$\mathbb{E}[P_1^m|\overline{P}] = \overline{p} - \sum_{i=1}^{n-m-k} \frac{1}{n-i}\zeta_i$$

$$P_1^m = \overline{p} - \sum_{i=1}^{n-m-k} \frac{1}{n-i}\zeta_i + \zeta_{n-k-m+1}$$

$$\mathbb{E}[P_2^m|\overline{P},P_1^m] = \overline{p} - \sum_{i=1}^{n-m-k+1} \frac{1}{n-i}\zeta_i$$

$$P_2^m = \overline{p} - \sum_{i=1}^{n-m-k} \frac{1}{n-i}\zeta_i + \zeta_{n-k-m+2}$$

$$\dots$$

$$\mathbb{E}[P_m^m|\overline{P},P_1^m,\dots,P_{m-1}^m] = \overline{p} - \sum_{i=1}^{n-k-1} \frac{1}{n-i}\zeta_i$$

$$P_m^m = \overline{p} - \sum_{i=1}^{n-k-1} \frac{1}{n-i}\zeta_i + \zeta_{n-k}$$

$$\sum_{i=1}^{m} P_m^m = (n-k)\overline{p} + \sum_{i=1}^{n-k} \frac{k}{n-i}\zeta_i - \sum_{i=1}^{n-m-k} \overline{P}_i$$

$$= m\overline{p} - \sum_{i=1}^{n-m-k} \frac{m}{n-i}\zeta_i + \sum_{i=n-m-k+1}^{n-k} \frac{k}{n-i}\zeta_i.$$

We can now use

$$\frac{1}{k}\sum_{i \in I_k} p_i = \frac{1}{k}\left(n\overline{p} - \sum_{i=1}^{n-m-k} \overline{P}_i - \sum_{i=1}^{m} P_i^m\right),$$

to substitute

$$\frac{1}{m} \sum_{i \in I_m} p_i - \frac{1}{k} \sum_{i \in I_k} p_i = \frac{1}{m} \sum_{i=1}^{m} P_i^m - \frac{1}{k} \left( n\overline{p} - \sum_{i=1}^{n-m-k} \overline{P}_i - \sum_{i=1}^{m} P_i^m \right)$$

$$= \frac{m+k}{mk} \sum_{i=1}^{m} P_i^m + \frac{1}{k} \sum_{i=1}^{n-m-k} \overline{P}_i - \frac{n}{k}\overline{p}$$

$$= \frac{m+k}{mk} \left( m\overline{p} - \sum_{i=1}^{n-m-k} \frac{m}{n-i}\zeta_i + \sum_{i=n-m-k+1}^{n-k} \frac{k}{n-i}\zeta_i \right)$$

$$+ \frac{1}{k} \left( (n-m-k)\overline{p} + \sum_{i=1}^{n-m-k} \frac{m+k}{n-i}\zeta_i \right) - \frac{n}{k}\overline{p}$$

$$= \sum_{i=n-m-k+1}^{n-k} \frac{m+k}{m(n-i)}\zeta_i$$

$$= \sum_{i=0}^{m-1} \frac{m+k}{m(k+i)}\zeta_{n-k-i}.$$

With these substitutions $Z$ can be written as a weighted sum of conditionally independent sub-Gaussian random variables:

$$Z = \sum_{i=0}^{m-1} \frac{m+k}{m(k+i)}\zeta_{n-k-i} + \frac{1}{m} \sum_{i \in I_m} \eta_i - \frac{1}{k} \sum_{i \in I_k} \eta_i.$$

Therefore $Z$ is according to Lemma 1 at least

$$\sqrt{\sum_{i=0}^{m-1} \left( \frac{m+k}{m(k+i)} \right)^2 + \sum_{i=1}^{m} \frac{1}{m^2} + \sum_{i=1}^{k} \frac{1}{k^2}} \leqslant \sqrt{\frac{3(m+k)}{mk}}$$

-sub-Gaussian.

The last step uses the inequality

$$\sum_{i=0}^{m-1} \frac{1}{(k+i)^2} = \int_0^m \frac{1}{(k+x)^2}\,dx + \sum_{i=0}^{m-1} \left( \frac{1}{(k+i)^2} - \int_{x=i}^{i+1} \frac{1}{(k+x)}\,dx \right)$$

$$= \frac{m}{(k+m)k} + \sum_{i=0}^{m-1} \frac{1}{(k+i)^2(k+i+1)}$$

$$\leqslant \frac{m}{(k+m)k} + \frac{1}{k+1} \sum_{i=0}^{m-1} \frac{1}{(k+i)^2}$$

$$\leqslant \frac{m(k+1)}{(k+m)k^2}$$

$$\leqslant \frac{2m}{(k+m)k}.$$

$\square$

# B   Proof of Theorem 1

With the Lemmas from the previous section, we can proof our main theorem.

*Proof of Theorem 1.* We follow the steps from the sketch.

**Step 1** We define the following shifted random variables.

$$\tilde{R}_t := R_t + \mu_t(a_*) - \mu_t(A_t)$$

$$\tilde{R}_s^i := \sum_{t \in T_s} \mathbb{I}\{A_t = a_i\}\tilde{R}_t$$

$$\Delta \tilde{D}_s^i := \frac{\tilde{R}_s^*}{N_s^*} - \frac{\tilde{R}_s^i}{N_s^i}$$

$$\tilde{D}_s(a_i) := \sum_{k=1}^s \min\{N_s^i, N_s^*\}\Delta \tilde{D}_k^i$$

$$\tilde{\Delta}_s(a_i) := \frac{\tilde{D}_s(a_i)}{N_{*,i}(s)}.$$

The reward functions satisfy $\mu_t(a_*) - \mu_t(a_t) > \Delta(a_t)$ for all $a_t$. Therefore $R_t > \tilde{R}_t - \Delta(A_t)$. So we can bound $\frac{D_{*,i}}{N_{*,i}} > \Delta(a_i) + \tilde{\Delta}_s(a_i)$ and $\frac{D_{i,*}}{N_{i,*}} < -\Delta(a_i) - \tilde{\Delta}_s(a_i)$.

Define the events

$$\mathcal{E}_s := \left\{\forall i : |\tilde{\Delta}_s(a_i)| \leqslant \sqrt{\frac{12\log(2Kf(N_{*,i})\delta_s^{-1})}{N_{*,i}}}\right\}, \quad \mathcal{F} := \bigcap_{s \geqslant 2}\mathcal{E}_s$$

and their complements $\overline{\mathcal{E}}_s, \overline{\mathcal{F}}$.

According to lemma 1, $\Delta \tilde{D}_s^i$ is $\sqrt{\frac{6}{\min\{N_s^*, N_s^i\}}}$-sub-Gaussian. So $\tilde{D}_s(a_i)$ is a sum of conditionally $\sigma_i$-sub-Gaussian random variables, such that $\sum_{i=1}^s \sigma_i^2 = 6N_{*,i}(s)$, Therefore we can apply Lemma 3. For both cases $\delta_s = \frac{1}{f(t_s)}$ and $\delta_s = \delta$, the probability never increases in time.

$$\mathbb{P}\left[\exists s' \geqslant s : \tilde{\Delta}_{s'}(a_i) \geqslant \sqrt{\frac{12\log(2Kf(\delta_{s'})N_{*,i})}{\delta_{s'}}}\right]$$

$$\leqslant \mathbb{P}\left[\exists s' \geqslant s : \tilde{D}_{s'}(a_i) \geqslant N_{*,i}\sqrt{\frac{12\log(2Kf(N_{*,i})\delta_s)}{N_{*,i}}}\right] \leqslant \frac{\delta_s}{2K}.$$

Using a union bound over $\pm\tilde{D}_s(a_i)$ for $a_i \in \mathcal{A}$, we get

$$\mathbb{P}[\overline{\mathcal{E}}_s] \leqslant \delta_s \text{ and } \mathbb{P}[\overline{\mathcal{F}}] \leqslant \delta_2.$$

**step 2** We split the number of pulls in two categories: those that appear in rounds where the confidence intervals hold, and those that appear in rounds where they fail: $N_t^{\mathcal{E}}(a_i) = \sum_{s'=1}^{s(t)} \mathbb{I}\{\mathcal{E}_s\}N_s^i$, $N_t^{\overline{\mathcal{E}}}(a_i) = \sum_{s'=1}^{s(t)} \mathbb{I}\{\overline{\mathcal{E}}_s\}N_s^i$.

$$N_t(a_i) \leqslant N_t^{\mathcal{E}}(a_i) + N_t^{\overline{\mathcal{E}}}(a_i)$$

$$\mathbb{E}[N_t^{\mathcal{E}}(a_i)] = \mathbb{P}[\overline{\mathcal{F}}]\mathbb{E}[N_t^{\mathcal{E}}(a_i)|\mathcal{F}] + \mathbb{P}[\overline{\mathcal{F}}]\mathbb{E}[N_t^{\mathcal{E}}(a_i)|\overline{\mathcal{F}}].$$

In the high probability case, we are with probability $1 - \delta$ in the event $\mathcal{F}$ and $N_t^{\overline{\mathcal{E}}}(a_i)$ is 0. In the setting of $\delta_s = f(t_s)^{-1}$, we can exclude the first round and start with $s = 2$ and $t_2 = M + 1$. This is because we do not use the confidence intervals in the first round.

$$\mathbb{E}[N_t^{\overline{\mathcal{E}}}(a_i)] \leqslant \sum_{s=2}^{\infty} \frac{t_{s+1} - t_s}{f(t_s)} \leqslant \sum_{s=1}^{\infty} \frac{M}{f(Ms)}$$

$$\leqslant \frac{M}{f(M)} + \sum_{s=2}^{\infty} \frac{M}{f(Ms)} \leqslant \frac{1}{2} + \sum_{s=1}^{\infty} \frac{1}{f(s)} \leqslant \frac{3}{2}$$

We use the fact that $\frac{1}{f(t_s)}$ is monotonically decreasing, so the expression gets minimized if all rounds are maximally long.

**Step 3:** bounding $\mathbb{E}[N_t^{\mathcal{E}}(a_i)|\mathcal{F}], \mathbb{E}[N_t^{\mathcal{E}}(a_i)|\overline{\mathcal{F}}]$

Let $s'$ be the last round at which the arm $a_i$ is not eliminated. We claim that $N_{i,*}$ at the beginning of round $s'$ must be surely smaller or equal to $\frac{48}{\Delta(a_i)^2}\left(\log(2K\delta_{s'}^{-1}) + 4\log(\frac{48\log(2K\delta_{s'}^{-1})}{\Delta(a_i)^2})\right)$. Assume the opposite holds, then according to Lemma 2 with $z = \frac{48}{\Delta(a_i)^2}$ and $y = \log(2K\delta_{s'}^{-1})$:

$$\frac{\frac{48}{\Delta(a_i)^2}(\log(f(N_{i,*}(s'))) + \log(2K\delta_{s'}^{-1}))}{N_{i,*}(s')} < 1 \qquad \Leftrightarrow \qquad \sqrt{\frac{12\log(2Kf(N_{*,i})\delta_s^{-1})}{N_{*,i}}} < \frac{1}{2}\Delta(a_i).$$

So we have that

$$\hat{\Delta}_{s'}^{LCB}(a_i) \geqslant \Delta(a_i) - 2\sqrt{\frac{12\log(2Kf(N_{*,i})\delta_s^{-1})}{N_{*,i}}} > 0,$$

and $a_i$ would have been excluded at the beginning of round $s'$, which is a contradiction.

Let $C(a_i)$ denote the number of plays of $a_i$ in round $s'$. Then for the different cases we have:

$$N_t^{\mathcal{E}}(a_i) - C(a_i) \leqslant \begin{cases} M \cdot N_{i,*}(s'), & \text{under the event } \overline{\mathcal{F}} \\ 2 \cdot N_{i,*}(s'), & \text{under the event } \mathcal{F} \\ N_{i,*}(s'), & \text{if } M_s = |\mathcal{A}_A| \end{cases}$$

$$\sum_{a \neq a_*} C(a) \leqslant \begin{cases} MK, & \text{under the event } \overline{\mathcal{F}} \\ M\log(K) + K, & \text{under the event } \mathcal{F} \\ K & \text{if } M_s = |\mathcal{A}_A| \end{cases}$$

The first case is trivial because each arm can only be played $M$ times in a single round and $\min\{N_s^i, N_s^*\} \geqslant 1$ in rounds with $\mathcal{E}_s$. The second case follows from the fact that $a_*$ is always in set $\mathcal{B}$ under the event $\mathcal{F}$. So $N_s^* \geqslant \max\{1, N_s^i - 1\}$ and $\min\{N_s^i, N_s^*\} \geqslant \frac{N_s^i}{2}$. The amount of pulls in a single round is naturally bounded by $\lceil \frac{M}{|\mathcal{B}|} \rceil \leqslant M$. Given that under the event $\mathcal{F}$, the set $\mathcal{B}$ never resets and the set $\mathcal{B}$ only decreases if an arm is eliminated, we can bound

$$\sum_{a_i \neq a_*} C(a_i) \leqslant \sum_{i=2}^{K}\lceil\frac{M}{i}\rceil \leqslant M\log(K) + K.$$

Finally the last case follows trivially because in the case of $M_s = |\mathcal{A}_A|$, we have $N_s^i = N_s^* = C(a_i) = 1$.

**Step 4:** combining everything

In the high probability case, we have with probability at least $1 - \delta$:

$$\begin{aligned} N_t(a_i) &\leqslant N_t^{\mathcal{E}}(a_i) + N_t^{\overline{\mathcal{E}}}(a_i) \\ &\leqslant 2N_{i,*}(s') + C(a_i) \\ &\leqslant \frac{96}{\Delta(a)^2}\left(\log(2K\delta^{-1}) + 4\log\left(\frac{48\log(2K\delta^{-1})}{\Delta(a)^2}\right)\right) + C(a_i) \end{aligned}$$

and also

$$\sum_{a \neq a_*} C(a) \leqslant M\log(K) + K.$$

If additionally $M_s = |\mathcal{A}_A|$, then the bound improves to

$$\begin{aligned} N_t(a_i) &\leqslant N_t^{\mathcal{E}}(a_i) + N_t^{\overline{\mathcal{E}}}(a_i) \\ &\leqslant N_{i,*}(s') + 1 \\ &\leqslant \frac{48}{\Delta(a)^2}\left(\log(2K\delta^{-1}) + 4\log\left(\frac{48\log(2K\delta^{-1})}{\Delta(a)^2}\right)\right) + 1. \end{aligned}$$

In the setting of $\delta_s = f(t_s)^{-1}$, we have

$$\mathbb{E}[N_t^{\mathcal{E}}(a_i) - C(a_i)] \leqslant 2N_{i,*}(s') + \frac{1}{f(M)}MN_{i,*}(s')$$

$$\leqslant \frac{120}{\Delta(a)^2}\left(\log(2Kt\log^2(t)) + 4\log\left(\frac{48\log(2Kt\log^2(t))}{\Delta(a)^2}\right)\right).$$

So

$$\mathbb{E}[N_t(a_i)] \leqslant \mathbb{E}[C(a) + N_t^{\overline{\mathcal{E}}}(a_i)] + \frac{120}{\Delta(a)^2}\left(\log(2Kt\log^2(t)) + 4\log\left(\frac{48\log(2Kt\log^2(t))}{\Delta(a)^2}\right)\right).$$

where

$$\sum_{a \neq a_*} \mathbb{E}[C(a) + N_t^{\overline{\mathcal{E}}}(a_i)] \leqslant M\log(K) + K + \frac{1}{f(M)}MK + \frac{3}{2}K$$

$$\leqslant M\log(K) + \frac{5}{2}K.$$

Finally if additionally $M_s = |\mathcal{A}_A|$, this bound improves to

$$\mathbb{E}[N_t(a_i)] \leqslant \mathbb{E}[N_t^{\overline{\mathcal{E}}}(a_i)] + N_{*,i}(s') + 1$$

$$\leqslant \frac{5}{3} + \frac{48}{\Delta(a)^2}\left(\log(2Kt\log^2(t)) + 4\log\left(\frac{48\log(2Kt\log^2(t))}{\Delta(a)^2}\right)\right).$$

$\square$

## C   Additional experiment

The winning probability is set to $0.95$. All sub-optimal arms are identical

Figure 4: Comparison with identical gaps of 0.9. The results are averaged over 20 repetitions of the experiment.