[Reviews · NeurIPS 2018]

Reviewer 1



This paper proposes a bandit model where the actions can be decomposed into the Cartesian product of atomic actions. It generalizes rank-1 bandits, relates their framework to combinatorial and contextual bandits and results in better bounds in the dueling bandit setting. Simple synthetic experiments show this framework indeed leads to better regret in practice. The theoretical bounds are reasonable, although some intuition explaining the bounds would be helpful. 1. Please consider moving the experiments to the main paper. 2. Please compare the performance of your algorithm (with possibly higher rank since you can incorporate it) to rank-1 bandits in real-world experiments proposed in that paper. This will make the paper much stronger. 3. Please formally characterize and compare the computational complexity of your algorithm to the elimination based algorithms. 4. Lines 127 - 130: Please explain this better. *** After Rebuttal *** I have gone through the other reviews and the author response. And my opinion remains unchanged.

Reviewer 2



This paper considers factored bandits, where the actions can be decomposed into a cartesian product of atomic arms. The paper is clear and well-written. The authors criticize the uniform playing of arms in rank-1 bandits. However, in TEA, the phase lengths are chosen to be equal to the size of the largest active set, and the arms are chosen uniformly at random from the active set in each dimension, so in expectation the arms along each dimension are played uniformly. Hence I wouldn't expect the sample complexity of TEA to be different from that of Rank1Elim for the rank-1 problem. In fact the authors comment that the sample complexity of TEA is higher than that of Rank1Elim, and argue that this is because TEA makes fewer assumptions. I would be interested in knowing if TEA can be modified to make the same strong assumptions as rank-1 bandits, and then compare it with rank1Elim. Other minor comments: 1) T in line 5 of Algorithm 1 should be t; similarly for Algorithm 2. *** After rebuttal *** My verdict is unchanged, and I understand that the main focus is finding the minimal assumptions under which learning is possible in the factored setup. However, I would like to see an analysis of TEA for the rank-1 case, that achieves similar or better regret bounds than the rank-1 paper.

Reviewer 3



The paper proposes to take a step back from several existing combinatorial bandit problems involving factored sets of actions (e.g. Rank-1 bandits). A meta algorithm based on sequential elimination is given and analyzed in two major settings: in general factored problems and in the dueling bandits setting. TL;DR: The whole algorithmic structure is extremely hard to parse… The improvement over Rank-1 bandits is not clear at all, and I am not an expert in Dueling Bandits to fully judge the impact of TEM on the constant” term in the analysis. Major comments: 1/ One recurrent inconsistency in the writing is that you define atomic arms at the beginning of section 2 and you barely use this vocabulary thereafter ! Arms and atomic arms are often called arms, especially in Definition 1 that actually defines gaps for atomic arms on each dimension. 1/ Algorithm description: For me, this is the main problem of this paper. The algorithm is extremely hard to parse for many reasons: many small inconsistencies (some are listed below) that do not help reading and a quite unclear description (l 130-137). Typically, I 134 : do you use comparisons between arms or atomic arms ? Algorithm 3 (function feedback) seems to do comparisons between atomic arms. - Algorithm 1 (and 2) : - M_s is not defined - This encapsulation of every step of your algorithm is actually more confusing than helping: basically you’re pulling all remaining atomic arm of each factor on each round. This is quite similar to Rank1Elim as far as I understand, and rather natural, but it took me quite some time making sure it was not more complicated than that. - Algorithm 3: If I understand well, you’re keeping track of all couples of atomic arms in each Module (so for each factor). That’s a quadratic memory complexity, typically of the order of LxK^2. Isn’t it an issue when K (number of atomic arm per factor) becomes large ? This is actually the interesting case of this type of structure I believe. - In function feedback, the indexing of N^i_s is inconsistent (sometimes N^s_i). - Using N_{i,j} and N^s_i for two different counters is quite confusing 2/ Link with Rank-1 bandits: TEA does not behave like rank1Elim but it could have been nice to explain better where is the improvement. Indeed, the rank-1 bandits is not a closed problem yet: there is still a gap between the lower bound of [Katariya et al.] and their algorithms (rank1Elim(KL)). If you think you are improving on this gap, it should be stated. If you are not, we would like you to tell us in which setting you are improving on the state of the art. I am really surprised by the empirical results. I see no reason why the rank1ElimKL would perform similarly as your policy, this is not discussed. 3/ Regret bound : It seems like your bound scales in 1/\Delta^2, which looks optimal but it is actually not. Typically, take the stochastic rank-1 bandits problem (which is a subclass of your setting). The deltas you defined are actually the smallest gaps while in [Katariya et al.] they consider the largest gaps (u_1v_1 - u_iv_1 = \Delta^U_i). They show a lower bound in 1/\Delta for such largest-gap definition. Their regret bound does not match their lower bound but it is still a bit tighter than yours as it scale as 1/(mean{u_1,…,u_K}) instead of 1/(min{u_1,…,u_K}). Minor comments: - l.22 Player The - It’s just my personal opinion but I think this paper tries to cover too wide of a scope. I am not an expert in dueling bandit but I know quite well the combinatorial / factored bandits literature in general. I think both contributions you make would be clearer if split in two separate papers. That would give you some space to explain better the behavior of your algorithm and where the improvement comes from. EDIT: Thank you for the detailed answers to my comments and remarks in the rebuttal. I raised my score a little bit because they were convincing :)